# Identification and Expression Analysis of Stilbene Synthase Genes in *Arachis hypogaea* in Response to Methyl Jasmonate and Salicylic Acid Induction

**DOI:** 10.3390/plants11131776

**Published:** 2022-07-05

**Authors:** Zuhra Qayyum, Fatima Noureen, Maryam Khan, Marrium Khan, Ghulam Haider, Faiza Munir, Alvina Gul, Rabia Amir

**Affiliations:** Department of Plant Biotechnology, Atta-ur-Rahman School of Applied Biosciences (ASAB), National University of Sciences and Technology (NUST), Islamabad 44000, Pakistan; zuhraqayyum39@gmail.com (Z.Q.); fatimanoureen2020@gmail.com (F.N.); maryamazamkhan@gmail.com (M.K.); marriumkhan96@gmail.com (M.K.); ghulam.haider@asab.nust.edu.pk (G.H.); faiza.munir@asab.nust.edu.pk (F.M.); alvina_gul@asab.nust.edu.pk (A.G.)

**Keywords:** stilbene synthase, methyl jasmonate, salicylic Acid, biotic stress, abiotic stress, stilbenoids

## Abstract

*Stilbene synthase* is an important enzyme of the phenylpropanoid pathway, regulating the production of several biologically active stilbenoids. These compounds have antioxidant, anti-inflammatory, and anti-cancer properties. However, the detailed characterization of *stilbene synthase* genes in *Arachis hypogaea* has not yet been performed. In this study, the comprehensive characterization of *stilbene synthase* genes in *A. hypogaea* was conducted, commencing with identification, phylogenetic analysis, and study of their expression in response to exogenous hormonal treatment. We identified and isolated five *AhSTSs* genes and recorded their expression pattern in peanut (BARD-479) in response to methyl jasmonate (MeJA) and salicylic acid (SA) treatment. The presence of Chal_sti_synt, ACP_syn_III, and FAE1_CUT1_rppA domains in all *AhSTSs* indicated their role in the biosynthesis of stilbene and lipid metabolism. Cis-regulatory element analysis indicated their role in light responsiveness, defense responses, regulation of seed development, plant growth, and development. Despite close structural and functional similarities, expression and correlational analysis suggested that these genes may have a specific role in peanut, as individual *AhSTS* exhibited differential expression upon hormonal treatment in a genotype dependent manner. Further studies on functional characterization involving the transcriptional regulation of *AhSTSs* can clearly explain the differential expression of stilbene synthase genes to hormonal treatment.

## 1. Introduction

Plants have developed certain physiological processes in response to the biotic and abiotic stress they encountered in their surroundings millions of years ago. Among these physiological processes is the synthesis of secondary metabolites [1]. One of these critical pathways in secondary metabolite production is the phenylpropanoid pathway, which controls the production of vital compounds such as lignin and flavonoids [2]. Flavonoids are important secondary metabolites produced in plants in response to abiotic and biotic stress. These compounds also regulate processes such as reproduction, cell physiology, and signaling [3].

A small group of substances in the phenylpropanoid group that are specifically involved in biotic and abiotic stress responses are known as phytoalexins [4]. The term phytoalexin is derived from Greek and means “warding off agents in plants” [5]. These compounds have low molecular weight, possess antimicrobial properties, and accumulate rapidly at the site of interaction with pathogens. Moreover, they also play a variety of protective roles, such as deterrents or repellent compounds, in plants [6]. Although phytoalexins have a wide chemical diversity throughout the plant kingdom, they are a relatively small group of molecules in peanuts that belong to the “stilbene family.” The stilbene derivatives in Fabaceae have a backbone of resveratrol [7] and antiradical compounds in plants [8]. It has antipathogenic [9], antioxidant [10], anticancer [11], and anti-inflammatory properties [12]. In the past few years, peanuts have gained considerable attention due to their high protein content and the presence of phytoalexin “resveratrol” in their leaves and pods. Moreover, the health benefits of resveratrol have been studied and established, as a result of which their pharmacological significance has been increased [13].

Stilbenes, along with flavonoids, form a class of compounds called “polyketides,” which represent the major group of phenylpropanoids. The compounds of the polyketide group are formed by the condensation of coumaric acid with three malonyl CoA subunits [14]. The synthesis of stilbene follows the phenylalanine/polymalonate pathway, also known as the shikimate pathway. The last step of the pathway is catalyzed by stilbene synthase (*STS*), an important enzyme in the pathway. *STS* belongs to the type III superfamily of polyketide synthase enzyme superfamilies. It catalyzes the condensation reaction of malonyl-CoA [15]. Simple stilbenes are synthesized in one step by the *STS* with the starter coenzyme A-esters of cinnamic acid derivative (*p*-coumaroyl-CoA). The p-coumaryl CoA is the starting material for the synthesis of resveratrol and cinnamoyl-CoA for the synthesis of pinosylvin, along with three malonyl-CoA units. The structure of *STS* is a homodimer and consists of two subunits of 40–45 kDa each [16]. *STS* is closely related to chalcone synthase (CHS), the key enzyme in flavonoid biosynthesis. The consensus sequence of *STS* and *CHS* from *Arachis hypogaea* shows 70% sequence identity [17]. The key reaction and substrate for *CHS* and *STS* are the same as the difference in the ring closure step. This step leads to two distinct products, chalcone and stilbenes, respectively [18].

*STS* genes have been cloned in several plant species such as peanut (*Arachis hypogaea*), Scots pine (*Pinus sylvestris*), eastern white pine (*Pinus strobus*), Japanese red pine (*Pinus densiflora*), grapevine (*Vitis vinifera*), and sorghum (*Sorghum bicolor*). *STS* occurs as a multigene family that is closely related to each other. In peanut and eastern white pine, two *STS* genes have been found [17,19]. On the other hand, Scots pine has five *STS* genes (*PST1, PST2, PST3, PST4,* and *PST5*) [20], while only one *STS* gene is present in sorghum [21]. Grapevine is the plant that contains the largest family of stilbene synthase genes, comprising approximately 48 *STS* genes [22]. Phytohormones play a vital role in plants by regulating their growth, development, and responses to various biotic and abiotic stresses through mediating several signaling pathways. The hormonal crosstalk during signaling regulates the distribution of secondary metabolites in stressed plants, maintaining a balance between plant growth and defense [23]. Salicylic acid and jasmonic acid derivatives, such as MeJA, act as effective elicitors, inducing the production of secondary metabolites in plants [24,25]. SA affects the growth and development of plants by regulating photosynthesis, transpiration, ion uptake, and transportation as well as affecting leaf morphology and chlorophyll structure [26]. SA also induces resistance against pathogens [27] and enhances the production of phenolic compounds [28]. MeJA regulates several biosynthetic pathways, including the phenyl-propanoid pathway in plants, by stimulating the catalytic activity of specific enzymes [29]. Moreover, it is an important abiotic elicitor, stimulating plant defense responses by regulating the production of secondary metabolites. It also boosts the production of phenolic compounds in cell cultures and plants as a whole [30].

The aim of the study was to explore the peanut genome to identify more members of the *STS* gene family and investigate the individual expression of *STS* genes to hormonal treatment. In this study, five *AhSTSs* genes were identified and reported for the first time from the peanut reference genome, along with their isolation and sequencing. Sequence and phylogenetic analysis unraveled their ancestry, conserved domains, motifs, cis-regulatory elements, physiochemical properties, active site residues, and 3D protein structure. The expression pattern of *AhSTSs* in response to hormonal treatment indicated their role in biotic and abiotic stresses in a genotype-dependent manner. This research provides a basis for unraveling the role of transcriptional factors in the differential expression of *AhSTSs* genes.

## 2. Results

### 2.1. Identification, Isolation and Sequence Analysis of AhSTSs Genes in Peanut

A total of nine sequences of *STS*, including five complete coding regions (CDS) and four partial CDS, were retrieved from the NCBI database (Appendix A). A BLAST search of these sequences in the Peanut DB and domain search resulted in the identification of five nucleotide sequences. Amplified sequences of *AhSTSs* in peanut showed 100% similarity to their retrieved sequences, respectively. Accession numbers of five *AhSTSs* genes identified in peanut along with selected sequence attributes have been provided in Appendix A. Nucleotide and amino acid alignment of the candidate genes through the Multialin tool showed greater than 90% sequence similarity for both nucleotide and amino acids in *AhSTS1, AhSTS2, AhSTS3, AhSTS4*, and *AhSTS6.* The alignment result indicated that *AhSTS1*, *AhSTS3,* and *AhSTS6* have close resemblance in terms of amino acid sequences (Appendix A). *Signal* P prediction confirmed the absence of signal peptide in all the genes. 

The identified full-length sequences of *AhSTSs* predicted that four of the proteins (*AhSTS1, AhSTS2, AhSTS3, AhSTS6*) have a length of 389 amino acids, whereas *AhSTS4* consists of 289 amino acids. Moreover, the result predicted that *AhSTSs* encode for thermostable proteins with a slight acidic character determined by their aliphatic index and isoelectric point as mentioned in (Appendix A). The subcellular localization indicated the presence of proteins in the cytoplasmic region. The extremely low values of GRAVY for predicted protein sequences of *AhSTSs* suggested that they form a stable interaction with water (Appendix A). The protein function is determined by the characteristic features of domain present in the protein structure. Therefore, SMART domain analysis identified three conserved domains in *AhSTSs* (Figure 1a). The N- and C-terminal domains comprising a Chalcone/Stilbene synthase domain and ACP synthase III domain. FAE1_CUT1_rppA domain was also detected in the center of the putative protein sequence, which predicted their role in lipid metabolism (Appendix A) Moreover, CDS region for *AhSTSs* identified through gene structure analysis has been elaborated in Figure 1b. The gene structure analysis showed the absence of introns in all sequences, which predicted that the genes consist of single intron-less coding regions. The results for the motif analysis identified six conserved motifs in the genes. Motif 5 was absent in *AhSTS4* protein while motifs 1,2,3,4 and 6 were conserved across the genes (Figure 1c). The presence of similar motifs in the protein sequences predicted the similarity of the biological function of proteins. All the amino acids in the motifs are not conserved, however, most of the amino acids are conserved (Figure 1d). Sequence logo analysis revealed the conservation of amino acids from the N- to C-terminal among the putative protein sequence of *AhSTS*s. The most observed amino acid residues predicted were valine, glycine, aspartic acid, serine, and phenylalanine. (Appendix A).

### 2.2. Phylogenetic Analysis of AhSTSs Genes in Peanut

Stilbenoids are a class of polyphenolic compounds found in some plant families. To gain insight into the phylogenetic relationship among *STS* from other plants, phylogenetic analysis was performed (Figure 2). The analysis showed that *AhSTSs* can be distributed into three monophyletic groups (I, II and, III). Group I was further subdivided into three subgroups A, B and, C. The phylogenetic tree showed multiple branches in group I. Whereas *AhSTSs* from peanut were distributed in group I, *AhSTS3, AhSTS4* and *AhSTS6* were clustered in subgroup A, and *AhSTS1* and *AhSTS2* were present in subgroup B. Moreover, several orthologous genes were found in group I. The subgroup C contained stilbene genes from *Vitis vinifera* and *Vitis riparia*. *AhSTS4* and *AhSTS6* showed a close relation to *STS2* from *Arachis duranensis*, while *AhSTS3* showed close resemblance to the stilbene synthase 3-like gene from *Arachis iapensis.* Furthermore, the results showed that *AhSTS4* and *AhSTS6* clustered together, hence, their function must be assessed to see whether they form a group of paralogous or orthologous genes. 

The phylogenetic analysis showed a close resemblance of the genes to each other, putting emphasis on the fact that *STS* belongs to the same group of multigene family. 

### 2.3. Promoter Sequence Analysis 

To understand the mechanism of the regulation of STS genes, putative *cis*-acting elements were identified using 1500 bp upstream promoter regions of *AhSTSs* in the peanut using Peanut DB (Figure 3 and Appendix A). A total of 213 *cis*-acting elements were predicted in the promoter region of *AhSTS*s. Stress-related responsive elements include light responsive elements (Box-4), (GATA-motif), (LAMP-element), (chs-CMA1a), (TCT-motif), (chs-Unit 1), (G-Box) and (G-box), elements involved in regulating seed development (RY-element), gibberellin-responsiveness (TATC-box), auxin responsive elements (TGA-elements), salicylic acid responsiveness (TCA-elements), MeJA responsiveness (CGTA-motif/TGACG-motif), and common *cis*-elements in the promoter and enhancer regions (CAAT). MeJA responsive elements (12, CGTA-motif/TGACG-motif) were predicted in the promoter region of *AhSTS2, AhSTS3, AhSTS4,* and *AhSTS6* but absent in *AhSTS1*. SA responsive elements (4, TCA-elements) were predicted in *AhSTS1, AhSTS3, AhSTS4*, and *AhSTS6* while absent in *AhSTS2.* It was revealed that the distribution of *cis*-regulatory elements varied in *AhSTS*s. The variation in *cis*-regulatory elements may be responsible for the differential expression of *AhSTSs* in response to hormonal treatment. That can be confirmed by further experimental proofs. Moreover, the prediction of anaerobic induction elements (ARE) suggested their role in abiotic responses needs to be confirmed through further experimentation.

### 2.4. Protein Structure Analysis of AhSTSs 

The secondary structure prediction analysis indicated the presence of alpha helices more than other secondary structures in the five enzymes (Appendix A). The percentage of alpha helices in *AhSTS1* was 43.19%, beta turns 5.66%, and random coils were 35.8%. In *AhSTS2,* the percentage of alpha helices was 44.73%, while the beta turns were 6.43%. A total of 33.98% were random coils. In *AhSTS3,* 44.73% alpha helical structure were present, with 5.14% beta turns, and 34.19% random coils. The percentage of alpha helices in *AhSTS4* was 43.19%, beta turns 5.66%, and random coils were 35.8%. Moreover, in *AhSTS6,* 44.73% alpha helical structure were present, with 5.14% beta turns, and 34.19% random coils. The 3-dimensional (3D) structure of *AhSTS* enzymes were predicted through homology modelling using SWISSMODEL (Appendix A). There were no disordered arrangements or states found for all five proteins. Moreover, the homology modelling analysis predicted that the *AhSTSs* forms a homodimer protein, consisting of two distinct α and ß chains. The predicted structure for the enzyme was validated through Ramachandran Plot analysis (Appendix A). The percentage of residues in the favored region was greater than 92% for the stilbene synthase proteins, enhancing confidence of the quality of the predicted structure. 

### 2.5. Enzyme–Substrate Interaction Analysis 

Enzyme–substrate interaction was performed to predict the activity of stilbene synthase enzyme. To determine the binding affinity of the substrate to the *AhSTS* enzyme, the best scoring poses with the least binding energy were examined and compared for *AhSTSs* enzymes, and their substrates and presented in Figure 4 and Figure 5. The amino acid residues lining the active site showed conventional hydrogen bonding, van der Waals interaction, carbon–hydrogen bonding, and unfavorable bumps with the substrate’s malonyl CoA and p-coumaroyl CoA. The active site residues of *AhSTSs* involved in interaction with malonyl CoA were Gly-305, Met-59, Asn-336, Thr-197, Gly-216, Gly-164, Lys-269, Val-271, Asp-270, Gly-149, Leu-150, Ser-250, Asp-249, Gly-376, Ser-250, Glu-251, Thr-197, Ser-338, Thr-132, Glu-255, His-303, Cys-164, Pro-272, Lys-269, Thr-63, Ile-59, Ala-216 Asn-139, Gly-305, Gly-163, Leu-267, Asn-268. Moreover, the active site residues of the five stilbene genes involved in interaction with p-coumaroyl CoA were Gly-216, Asn-336, Cys-264, Lys-182, Glu-285, Lys-155, Lys55, Val-210, Ile-59, Lys-269, Gly-163, Asn-193, Ala-216, Gly-306 Arg-307, Leu-214, Arg-62, Lys-269, Asn-268, Tyr-266, Cys-164, Gly-255, Thr-264, Asn-193, Asp-217, Ala-216, Pro-89, Glu-260, Ala-93, Asn-80, Asp-96, Arg-100, Pro-138, His-251, and Glu-143. The bond length was calculated using discovery studio tool. As it formed a favorable interaction with malonyl-CoA and p-coumaroyl CoA, it predicts that the enzyme utilizes these compounds as substrates during the stilbene biosynthetic pathway. 

### 2.6. Expression Characterization of AhSTSs in Response to MeJA and SA 

Differential expression of *AhSTSs* genes have been observed in the leaves of peanut (BARD-479) in response to MeJA and SA hormonal treatment (Figure 6). All *AhSTSs* genes in BARD-479 were significantly up-regulated under the influence of MeJA treatment at all-time points except *AhSTS*2 that showed a 3.6 mean fold increase (*p*-value = 0.028) only at 24 h. A higher expression of *AhSTSs* was observed in the plants treated with hormones as compared to control plants. The results were in accordance with those found by Vannozzi et al [31] The expression of genes was higher during the initial few hours of hormonal treatment, and this declined with time. Some of the genes also showed enhanced expression after 6 h of hormonal treatment that can be linked to the transcriptional regulation of genes. The correlation analysis of *AhSTSs* with respect to time in BARD-479 revealed a strong positive correlation of *AhSTS*1 (*r* = 0.8905), *AhSTS*4 (*r* = 0.9622) and moderate correlation with *AhSTS*2 (*r* = 0.7479) under MeJA treatment. 

The maximum expression of *AhSTS1* was observed at 0.5 h after SA treatment in BARD-479, which gradually declined until 6 h followed by gradual increase until 24 h. For *AhSTS2,* the expression in BARD-479 was significantly up-regulated at 0.5 h, 6 h, and 24 h after SA treatment. Expression of *AhSTS3* in BARD-479 was up-regulated at 0.5 h and 6 h after SA treatment. *AhSTS4* showed a gradual increase in its expression in BARD-479 till 8 h. *AhSTS6* showed up-regulation at 6 h after SA treatment, while in BARD-479 up-regulation was recorded at 6 h and 8 h. A moderate degree of negative correlation has been observed for *AhSTS*1 (*r* = −0.4962), *AhSTS*2 (*r* = −0.5012), *AhSTS*3 (*r* = −0.4976) in BARD-479 with respect to time, while little or no association has been observed for *AhSTS4* and *AhSTS6* in BARD-479 after SA treatment (Appendix A). 

Pearson’s correlation coefficient revealed that MeJA affected expression of *AhSTS2*, *AhSTS3* and *AhSTS4* was negatively correlated to SA affected expression of *AhSTS2*, *AhSTS3* and *AhSTS4* in peanut (Appendix A). Under SA treatment, the expression of *AhSTS2* is positively correlated with the expression of *AhSTS3* in peanut. The expression of *AhSTS6* was negatively correlated with the expression of *AhSTS1*, *AhSTS2,* and *AhSTS3* in peanut (Appendix A). This showed that MeJA and SA works in antagonistic order. 

## 3. Discussion

Stilbenes are well known phytoalexins, regulating plant defense responses against biotic and abiotic stresses in several plants [32]. *STS* is one of the vital enzymes of phenylpropanoid pathway that catalyze the biosynthesis of plant stilbenes [33]. Owing to their physiological and pharmaceutical significance, stilbenes have been subjected to intensive research. Peanut stilbenes are considered as one of the major sustaining factors of the plant’s resistance to diseases and have proven to be beneficial for human health [34,35]. Genome-wide identification and characterization of *stilbene synthase* genes have been performed in some plants but the comprehensive characterization of *STS* in peanut has not been reported. In the present study, the five *AhSTSs* genes were identified and isolated from peanut, followed by molecular characterization. Although the number of *STS* genes in peanut is less than compared to other plants in higher taxonomic status, it shows that *STS* belongs to a multigene family with significant importance in peanut. Moreover, the role of *AhSTSs* against biotic and abiotic stress was evaluated through the expression analysis under the influence of MeJA and SA treatment in peanut (BARD-479). 

The phylogenetic analysis showed the independent evolution of *AhSTSs* in *Arachis hypogaea* from other plant families and close sequence similarity to parent species, i.e., *A. iapensis* and *A. duranensis*. According to phylogeny, *AhSTS1, AhSTS2* and, *AhSTS3* evolved earlier while *AhSTS4* and *AhSTS6* evolved later. Furthermore, sequence analysis predicted that *AhSTS1, AhSTS3,* and *AhSTS6* showed greater sequence similarity as compared to *AhSTS4* and *AhSTS2.* Based on phylogenetic analysis, *STS*s can be distributed into three monophyletic groups (I, II and, III). Among, the stilbenes, *AhSTSs* were clustered in Group I, which was further subdivided into three subgroups A, B, and C based on their close evolutionary relation. Moreover, based on sequence similarities, *AhSTS3, AhSTS4,* and *AhSTS6* were clustered in subgroup A, and *AhSTS1* and *AhSTS2* were present in subgroup B. Thus, it is predicted that *AhSTS*2 might be closely related to the gene of origin, while *AhSTS4* and *AhSTS6* are recently evolved genes, however, *AhSTS4* is more diverse than *AhSTS6*. The analysis also predicted that the *AhSTS* genes belong to a multigene family closely related to each other based on their distance from each other in the phylogenetic tree.

The presence of Chal_sti_synt and ACP_syn_III domains in all the *AhSTSs* predicted their role in the biosynthesis of stilbenes as they are involved in condensation of *p-*coumaroyl CoA [18] and malonyl CoA to resveratrol, while the presence of the FAE1_CUT1_RppA domain predicted their role in lipid metabolism [24]. Gene structure and protein motif analysis predicted the sequence conservation in peanut stilbene genes. The results for motif analysis identified six conserved motifs in the genes. Motif 5 was absent in *AhSTS4* protein while motifs 1,2,3,4 and 6 were conserved, which predicted that *AhSTS4* is advanced among all *AhSTSs*. Secondary structure predicted more alpha helices than other secondary structures in *AhSTS*s, while homology modelling analysis predicted that the *AhSTSs* form a homodimer protein consisting of two distinct chains. SeqLogo analysis demonstrated the conservation of amino acids from N- to C-terminal, among *AhSTSs* proteins, and showed a slightly acidic character of protein due to the presence of aspartic acid and glutamic acid. Moreover, this analysis also predicted that the most observed amino acid residues were valine, glycine, aspartic acid, serine, and phenylalanine.

The docking of protein with substrates can be used to predict the binding affinity of enzyme to possible substrates. Our docking results predicted that *AhSTS* enzymes displayed high binding affinity for malonyl CoA and *p*-coumaroyl CoA, which are initial substrate for stilbene synthesis [36] as all *AhSTSs* formed favorable interactions with both substrates, respectively, indicating their role in stilbene biosynthesis. Promoter region analysis predicted the role of *AhSTSs* in light responsiveness, defense responses, regulation of seed development, regulation plant growth, and development. Moreover, presence of hormone responsive *cis*-regulatory elements predicted their responsiveness for gibberellic acid, auxin, ABA, SA, and MeJA, which can be confirmed through further analysis.

The expression patterns of the gene during various environmental conditions helps to determine the role of the genes in plant growth and development. So, we examined the expression of *stilbene synthase* in response to MeJA and SA treatment. Despite close structural and functional similarities, expression and correlational analysis suggested that these genes might have a specific role in peanut, as individual *AhSTS* exhibited differential expression upon hormonal treatment in a genotype dependent manner. MeJA mostly up-regulated the expression of *AhSTSs* genes ascompared to SA, which suggested these genes mostly participate in abiotic responses because MeJA is an essential regulator of abiotic stress responses [37]. On the contrary, SA, a crucial regulator of biotic stress responses [37], down-regulated the expression of *AhSTSs* at various time points in peanut but time-dependent up-regulation has also been observed in all *AhSTS*s, which indicated that the expression of *AhSTSs* might be tightly regulated during biotic stress. Moreover, correlation analysis revealed that the expression of *AhSTS*1, *AhSTS*4, *AhSTS*2, and *AhSTS6* is positively correlated with time after MEJA treatment while no such correlation was observed in case of SA treatment. On the other hand, moderate degree negative correlation has been observed for *AhSTS*1, *AhSTS*2, and *AhSTS*3 in BARD-479 with respect to time after SA treatment, which indicated that *AhSTSs* are early responsive genes against biotic stresses.

In addition, Pearson’s correlation coefficient indicated the antagonistic regulatory behavior of MeJA and SA, as moderate level negative correlations have been observed when the MeJA-affected expression of *AhSTSs* is compared to the SA-affected expression of *AhSTS*s. *AhSTSs* exhibited differential expression in genotypes under both hormonal treatments, which was also confirmed by the study in [31]. Additionally, correlation among the expression of *AhSTSs* in peanut suggested a coordinated response of *AhSTSs* in genotype and stimuli-dependent manner. Diversity in *AhSTSs* response to hormonal treatment in different genotypes also provides a hint for co-regulation to create specificity in the response. Further studies on the regulation and functional analysis of *AhSTSs* may reveal more interesting knowledge of these genes, reveal their use as a potential target for metabolic engineering for higher production of stilbenes at industrial scale, and help to develop disease-resistant crop plants.

## 4. Materials and Methods

### 4.1. Identification of Stilbene Synthase Genes in Peanut

National Center for Biotechnology Information (NCBI) database was searched for *Stilbene synthase* (*STS*) of peanut. The retrieved sequences were further observed in the SMART database (http://smart.emblheidelberg.de/, accessed on 20 August 2019) to confirm the presence of *STS* domain [38]. A total of 9 sequences of *STS*, including 5 complete coding region (CDS) and 4 partial CDS, were retrieved from NCBI database. FASTA sequence of all the nucleotide sequences from NCBI were BLAST in Peanut DB (http://bioinfolab.muohio.edu/txid3818v1, accessed on 20 August 2019) [39], which resulted in long list of contigs. Among all these contigs, contigs 2394 and 20,452 showed the presence of complete conserved domains for *STS*, which were further used to complete partial contigs through alignment. A total of 5 candidate genes were identified in peanut. Sequence specific primers were designed to confirm these sequences in peanut genome. Primer sequences have been given in Appendix A for the reference. 

### 4.2. Sequence Analysis 

The nucleotide sequences of *AhSTSs* were further analyzed to predict intron/exon junction, *cis*-regulatory elements, and motifs. Gene structure display server (http://gsds.gao-lab.org/, accessed on 20 July 2020) was used to predict CDS and gene structure of *AhSTSs* display using default parameters [40]. ExPASY translate tool was used to get the open reading frame (ORF) of nucleotide sequence of *AhSTS*s. ExPASY (https://web.expasy.org/protparam/, accessed on 20 July 2020) protparam tool was used to calculate the molecular weight, amino acid length, isoelectric point (pI), aliphatic index (AI), GRAVY, and half-life of proteins [41]. The transmembrane region for *AhSTS1, AhSTS2, AhSTS3, AhSTS4,* and *AhSTS6* was predicted using SOSUIsignal (http://harrier.nagahama-i-bio.ac.jp/sosui/sosuisignal/sosuisignal_submit.html, accessed on 20 July 2020), respectively. SignalP 5.0 server was used to predict the presence of signal peptide cleavage sites for *AhSTS1, AhSTS2* and *AhSTS3, AhSTS4, AhSTS6* under the default settings, respectively.

Peanut database was used to extract the upstream promoter region of the genes comprising of 1500 bp. The promoter region was analyzed in PlantCare database (http://bioinformatics.psb.ugent.be/webtools/plantcare/html/, accessed on 21 February 2021) for predicting cis-regulatory elements. MEME software (http://meme-suite.org/, accessed on 21 February 2021) was used to predict the motifs present at default parameters. TB tools software was used to draw the motif pattern. 

### 4.3. Phylogenetic Analysis

Nucleotide sequence of *AhSTSs* were then aligned with sequences from *A. duranensis, A. iapensis, O. sativa, M. trunculata, V. vinifera, V. riparia*, and *A. thaliana* using MUSCLE alignment at default settings. An unrooted phylogenetic tree was constructed using the Neighbor-joining method at a bootstrap value of 1000 in MEGA X. Seq2 logo was used to generate the position specificity of amino acids in *AhSTSs* under default parameters through sequence alignment (http://www.cbs.dtu.dk/biotools/Seq2Logo, accessed on 1 July 2021).

### 4.4. Prediction of Protein Structure 

PSI-blast based secondary structure prediction (PSIPRED) and Chou and Fasman secondary structure prediction (CFSSP) server [42] were used to predict the alpha helices and beta sheets in the candidate proteins. SWISS-MODEL was used for predicting three-dimensional structure of stilbene synthase proteins for the five genes, based on homology modeling. The detailed structure of proteins was studied in Discovery studio. Individual secondary structures, such as alpha helices, beta sheets, coils, and protein chains, were identified in proteins. The protein quality was independently evaluated and verified for *AhSTS1, AhSTS2, AhSTS3**, AhSTS4* and *AhSTS6* through ERRAT, (https://servicesn.mbi.ucla.edu/ERRAT/, accessed on 3 June 2020), QMEAN (https://swissmodel.expasy.org/qmean/, accessed on 3 June 2020) [43], and Ramachandran plot (http://mordred.bioc.cam.ac.uk/~rapper/rampage.php, accessed on 16 June 2020) [44]. The protein models were submitted to COFACTOR (http://zhanglab.ccmb.med.umich.edu/COFACTOR/, accessed on 23 August 2020) for the prediction of active site residues [45]. 

### 4.5. Enzyme–Substrate Interaction 

Interaction between *AhSTS* enzyme and their substrate was predicted using molecular docking in pyrx with Autodock Vina algorithm. Based on literature, p-coumaroyl CoA and malonyl CoA were selected as the ligand of *AhSTSs* enzyme for analysis. The interaction of the ligand with active site residues and bond length was analyzed through Discovery studio. 

### 4.6. Plant Material and Hormonal Treatment

The seeds of registered commercial peanut variety (BARD-479) developed by Oilseed department of National Agricultural Research Centre (NARC), Pakistan were obtained after permission from authorized person. All the experiments were performed according to the guidelines provided by Coordinated Framework for Regulation of Biotechnology and institutional review board (IRB), ASAB, NUST, Pakistan. Seeds were de-husked and surface sterilized using 70% ethanol followed by washing with autoclaved distilled water. To break seed dormancy, seeds were kept at 4 °C for two days. Seeds were kept in dark for germination in petri plate provided with moist filter paper. Equal sized seedlings were transferred in autoclaved mixture of soil and peat moss in 1:1 ratio. Seedlings were grown at temperature of 28 °C and photoperiod of 16/8 h. Two-week-old plantlets at their trifoliate stage were given hormonal treatment (100 µM solution of MeJA and 1 mM solution of SA, respectively) through Foliar Spray Method The concentration of MeJA and SA were based on results shown by Vannozi et al. [31] and Souri et al. [46], respectively. Working solution of 100 µM MeJA was prepared by adding 3.3 µL of MeJA in 150 mL distilled water that contained 2 mL *v*/*v* absolute ethanol. The 1 mM SA stock solution was prepared by dissolving 27.62 mg of SA in warm 200 mL dH20 with 10 mL absolute ethanol. Hormonal solution was then sprayed two times individually on the leaves of the 15 peanut plants for each hormone. Leaf samples of treated and control plants were collected at 0 h, 0.5 h, 3 h, 6 h, 8 h, 12 h, and 24 h in an Eppendorf tube followed by immediate freezing in liquid nitrogen. Control plants were used to normalize expression of genes. Treatment at 0 h had been selected to measure the variation of expression over the time. Samples were stored in −80 °C for further experiments. 

### 4.7. RNA Isolation, cDNA Synthesis and RT-PCR

Total RNA was extracted from leaves sample of both control and treated peanut plants (BARD-479) through the Trizol reagent method [47]. Integrity and purity of RNA was tested by agarose gel electrophoresis and NanoDrop™ 2000/2000 c Spectrophotometers. An amount of 1 µg of total RNA was converted into cDNA using Thermo Scientific™ RevertAid First Strand cDNA Synthesis Kit using oligo dT primers. Reverse transcriptase-polymerase chain reaction (RT-PCR) was performed to confirm the presence of candidate genes in the peanut genome.

### 4.8. Cloning and Sanger Sequencing 

Amplicons were cleaned with EXoSAP-IT ^TM^ PCR product Cleanup Reagent (Thermo ScientificWILM, Wilmington, NC, USA) and cloned in DH5α of *E. coli* through electroporation. Clones were sequenced to confirm amplified gene product. Obtained sequences were submitted in NCBI to get accession number and termed as *AhSTS1, AhSTS2, AhSTS3, AhSTS4,* and *AhSTS6.*

### 4.9. Relative Gene Expression Analysis

Relative gene expression of *AhSTSs* was determined in the leaves sample of selected peanut variety at 0.5 h, 3 h, 6 h, 8 h, and 24 h in relation to control plant samples. The 7500 Fast Real-Time PCR system (Applied Biosystem, WLM, MA, USA) was used for real-time PCR analysis, according to instructions given in Maxima SYBR Green/ROX qPCR Master Mix by Thermo Fisher Scientific. Fold changes were calculated using Livak method (2-ΔΔCt method) [48]. Actin was used as housekeeping gene. Primer sequences used in the analysis has been given in Appendix A. Each experiment was replicated three times by taking individual leaves from three plants at the different time points. 

### 4.10. Statistical Analysis 

Microsoft Excel 365 and GraphPad Prism^®^ version 9 was used to organize and analyze data. Two-way ANOVA (analysis of variance) was performed with Bonferroni post hoc test to measure significant variation in control vs. treated samples. Correlation analysis was performed using Pearson’s correlation coefficient. *p*-value 0.05 was considered to be statistically significant. 

## 5. Conclusions

In the present study, we have identified five *AhSTS* genes in peanut. These *AhSTS* genes were subdivided into three subgroups based on sequence identity and evolutionary relativeness. Several cis-regulatory elements were identified, which showed their role in light responsiveness, defense responses, the regulation of seed development, plant growth and development. Most of the genes were differentially expressed in response to MeJA and SA treatment in a time-dependent manner, indicating the regulation of plant defense responses. Our findings pave way for exploring the potential of *AhSTS* genes to improve the resistance of peanut to biotic and abiotic stress through genetic engineering. The study on transcriptional regulation of these genes would help us better understand the differential expression and how to get increased stilbene production in plants.

## Figures and Tables

**Figure 1 plants-11-01776-f001:**
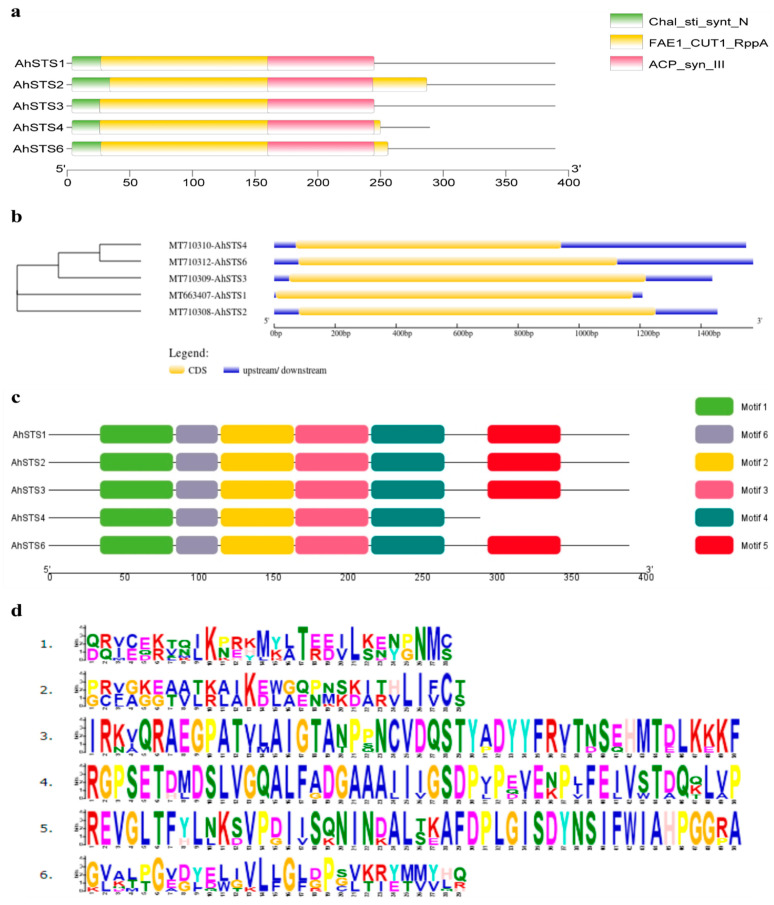
Molecular characterization of *Stilbene synthase* (*AhSTS*s) genes in peanut. (**a**) Domain analysis of putative amino acid sequence of *AhSTS*s, indicating Chalcone/stilbene synthase domain at the C and N-terminal, ACP synthase III domain and FAE1_CUT1_RppA in the center. (**b**) Gene structure analysis showing the coding region (CDS) for *AhSTSs* where yellow color is representing CDS. (**c**) 6 conserved motifs of *AhSTSs* observed across all genes where motif 5 was absent in *AhSTS4* (**d**) The amino acid sequence from motif 1 to motif 6. The conserved amino acids are represented by larger symbols while the smaller and stacked amino acids are not conserved.

**Figure 2 plants-11-01776-f002:**
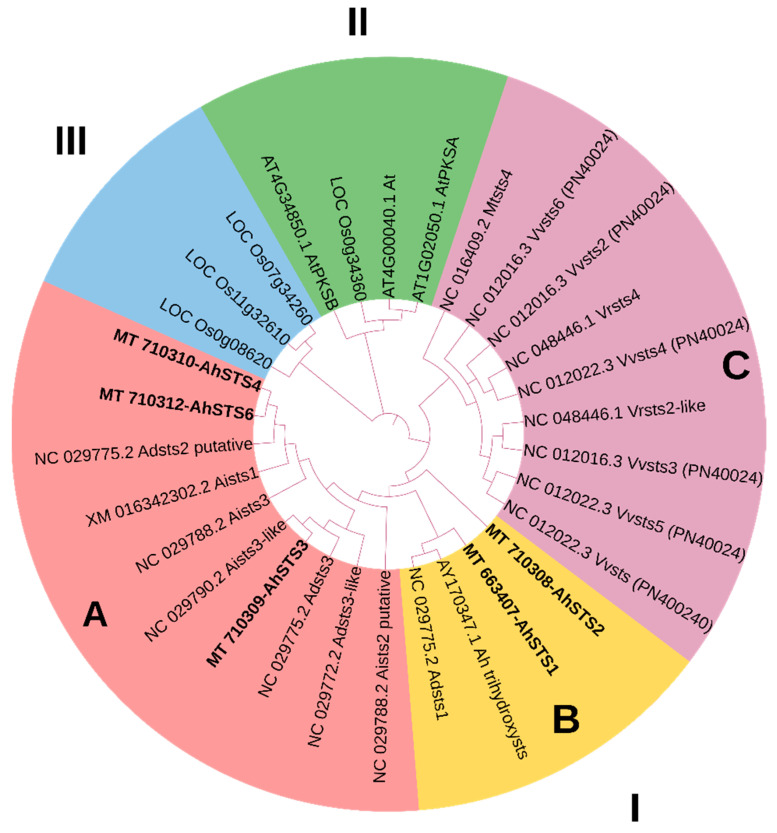
Phylogenetic analysis of *AhSTSs* through Neighbor Joining method using bootstrap with 1000 replicates indicated three monophyletic groups. *AhSTSs* are highlighted in bold, whereas pink, rust, and light violet represent group I, green represents group II, and blue represent group III.

**Figure 3 plants-11-01776-f003:**
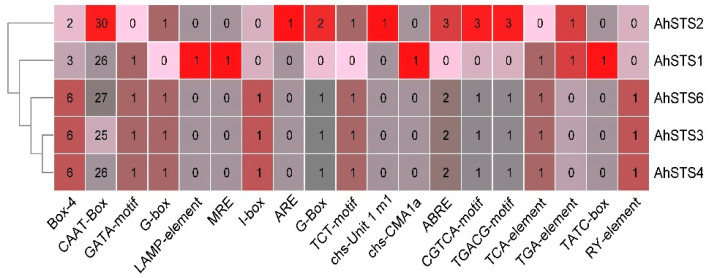
Heat map of cis-regulatory elements in promoter region of *AhSTSs*. Numbers in the heatmap are showing the predicted number of specific cis-regulatory elements in a particular gene.

**Figure 4 plants-11-01776-f004:**
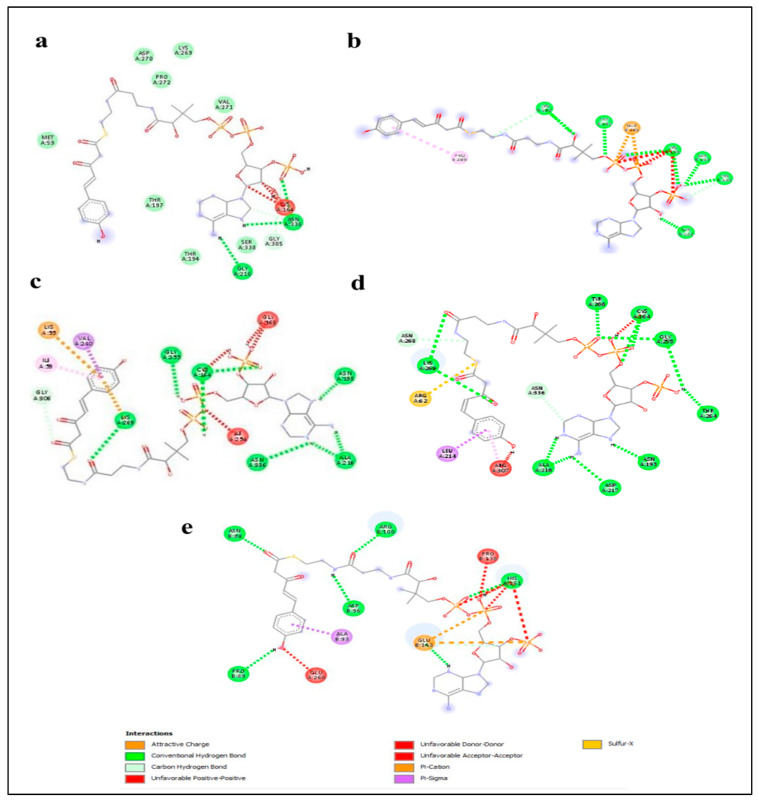
Interaction of active site residues of peanut *Stilbene synthase* (*STS*s) enzymes with p-coumaroyl CoA as substrate (**a**) *AhSTS1* (**b**) *AhSTS2* (**c**) *AhSTS3* (**d**) *AhSTS4* (**e**) *AhSTS6*, respectively. The higher the number of hydrogen bonds, the more stable interaction.

**Figure 5 plants-11-01776-f005:**
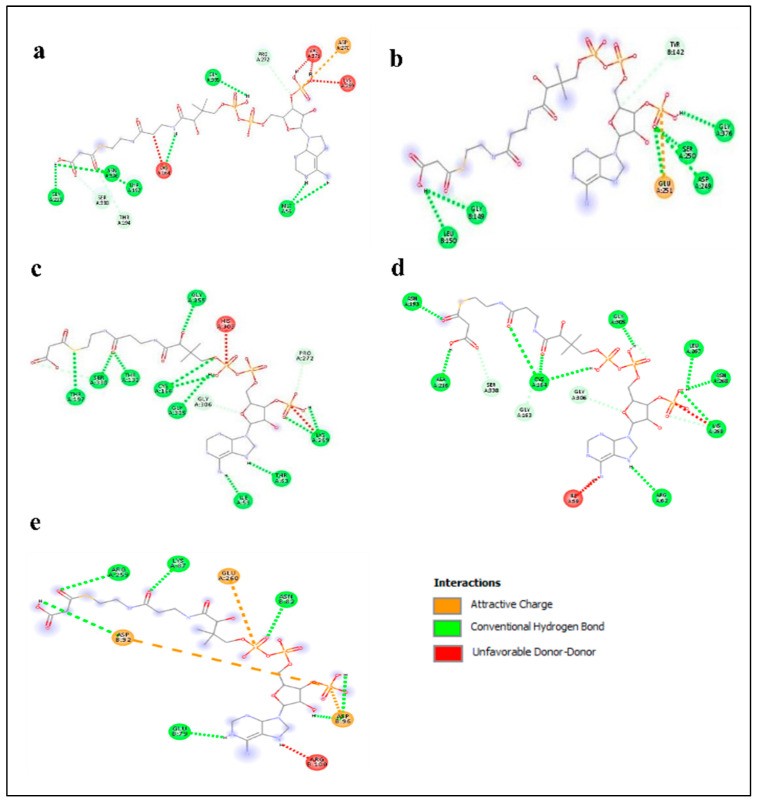
The interaction of (**a**) *AhSTS1* (**b**) *AhSTS2* (**c**) *AhSTS3* (**d**) *AhSTS4* and (**e**) *AhSTS6* with malonyl CoA. The higher the number of conventional hydrogen bonds, the more stable the interaction is that formed among enzymes and substrates.

**Figure 6 plants-11-01776-f006:**
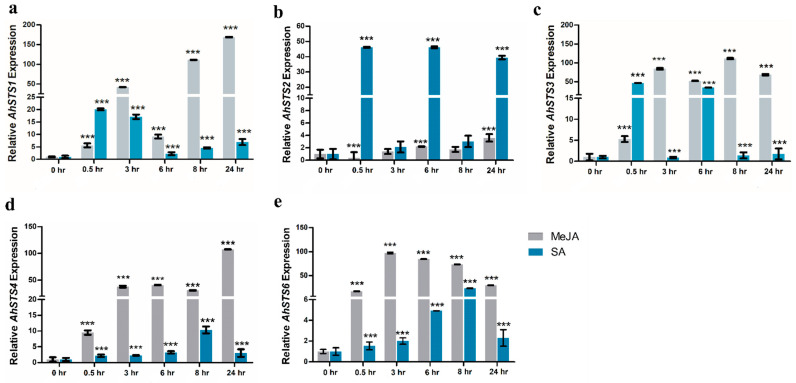
Methyl jasmonate and SA affected expression analysis of (**a**) *AhSTS1* (**b**) *AhSTS2*, (**c**) *AhSTS3,* (**d**) *AhSTS4* (**e**) *AhSTS6* in the leaves of peanut plant. The expression of *AhSTSs* were recorded at different time intervals, i.e., 0-h, 0.5-h, 3-h, 6-h, 8-h, and 24-h in peanut leaves. The asterisk signs on the bar graph shows *p*-value < 0.001.

## Data Availability

Not applicable.

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
