# Peer review of "Identification and Expression Analysis of Stilbene Synthase Genes in Arachis hypogaea in Response to Methyl Jasmonate and Salicylic Acid Induction"

_plants, 2022, doi:10.3390/plants11131776_

Round 1

Reviewer 1 Report

Major:

1) The main topic of the criticism is authors did not present data about stilbene content in the peanuts Arachis hypogaea leaves in control conditions and after methyl jasmonate (MeJa) or salicylic acid (SA) treatment.

2) In my opinion, the authors should provide information about all the STS sequences found in peanuts A. hypogaea, as an additional supplementary table instead of present supplementary Table 1. Now supplementary Table 1 is the repetition of the Table 1 in the main manuscript (Ms) text.

3) Authors should explain used concentrations for MeJa or SA treatment (100mM solution of MeJA and 1mM solution of SA, respectively). Compared those concentrations with other concentrations used by other scientist in the previously published papers.

4) Authors should improve the statistical treatment, e.g. why so high standard errors (S.E.) in the Fig. 3 (n, Golden, 0.5 and 8 h) if authors used S.E.?

5) Also, authors should increase the quality of the Figures 1 and 2.

6) Authors should explain in details the method used for plants treatment. “Line 446: “Foliar Spray Method”” this explanation is not enough.

7) Authors should improve all legends for figures, e.g. explain all used abbreviations, in Fig. 3 include a short description of what and how were treated by used substances (by MeJa and SA). Also, I advise authors to remove the description of the ANOVA values, present them in the supplementary materials.

8) Line 473: “Actin was used as housekeeping gene”, Authors should use at least 2 internal controls in real time PCR

Minor:

9) Line 17: correct “Arachis hypogaea” to “A. hypogaea”.

10) Line 80: “STS” in italic.

11) Line 248: correct “30.2) (f) AhSTS2” to “30.2) (f), AhSTS2”.

12) Line 255: correct “35.5) (m) and” to “35.5) (n) and”.

13) Figure 3: authors should present Fig. 3a,b,c,d,e as the supplementary material.

14) Table 3,4, and 5: authors should present those tables as the supplementary material.

Author Response

Response to Reviewer 1 Comments

Point 1: The main topic of the criticism is authors did not present data about stilbene content in the peanuts Arachis hypogaea leaves in control conditions and after methyl jasmonate (MeJa) or salicylic acid (SA) treatment.

Response: Yes, we agree that quantification of stilbene content should be conducted to correlate expression of stilbene gene with the stilbene content. However, due to lack of resources and facilities we could not perform quantification of stilbene. So, our study is limited to the expression of stilbene synthase genes to MeJA and SA treatment.

Point 2: In my opinion, the authors should provide information about all the STS sequences found in peanuts A. hypogaea, as an additional supplementary table instead of present supplementary Table 1. Now supplementary Table 1 is the repetition of the Table 1 in the main manuscript (Ms) text.

Response: Required information regarding all the sequences of stilbene in Arachis hypogaea has been added in the supplementary file Table S1.

Point 3: Authors should explain used concentrations for MeJa or SA treatment (100mM solution of MeJA and 1mM solution of SA, respectively). Compared those concentrations with other concentrations used by other scientist in the previously published papers.

Response: It has been updated in the revised manuscript. Please refer to the line 427 – 429 in revised manuscript.

Point 4: Authors should improve the statistical treatment, e.g. why so high standard errors (S.E.) in the Figure 3 (n, Golden, 0.5 and 8 h) if authors used S.E.?

Response: It has been updated in the revised manuscript.

Point 5: Also, authors should increase the quality of the Figures 1 and 2.

Response: The quality of figures have been improved in the revised manuscript. Please refer to the Figures 1, 2, 3 and 4.

Point 6: Authors should explain in detail the method used for plants treatment. “Line 446: “Foliar Spray Method”” this explanation is not enough.

Response: It has been updated in the revised manuscript. Please refer to the lines 429 – 432 in the revised manuscript.

Point 7: Authors should improve all legends for figures, e.g. explain all used abbreviations, in Figure 3 include a short description of what and how were treated by used substances (by MeJa and SA. Also, I advise authors to remove the description of the ANOVA values, present them in the supplementary materials.

Response: It has been updated in the revised manuscript. Please refer to the lines 254 – 256.

Point 18 Line 473: “Actin was used as housekeeping gene”, Authors should use at least 2 internal controls in real time PCR

Response: Two internal control are not needed here. The actin was used as per available literature and was worth to go for the expression analysis of MeJA and SA.

Minor:

Point 9: Line 17: correct “Arachis hypogaea” to “A. hypogaea”.

Response: It has been updated in the revised manuscript. Please refer to the line 17 in the revised manuscript.

Point 10: Line 80: “STS” in italic.

Response: It has been updated in the revised manuscript. Please refer to the line 80 in the revised manuscript.

Point 11:  Line 248: correct “30.2) (f) AhSTS2” to “30.2) (f), AhSTS2”.

Response: It has been updated in the revised manuscript. Please refer to the line 234 – 236 in the revised manuscript.

Point 12:  Line 255: correct “35.5) (m) and” to “35.5) (n) and”.

Response: It has been updated in the revised manuscript. The image has been updated with the captions.

Point 13: Figure 3: authors should present Figures 3a,b,c,d,e as the supplementary material.

Response: It has been updated in the revised manuscript. Please refer to the supplementary file Figures S10-14.

Point 14: Tables 3, 4 and 5: authors should present those tables as the supplementary material.

Response: It has been updated in the revised manuscript. Please refer to the supplementary file Table S6.

Reviewer 2 Report

This manuscript presents the identification and an initial characterization of the family of stilbene synthase genes in peanut. The study uses qPCR expression analysis and multiple in silico approaches, however I question whether all the in silico analyses are informative and necessary. A general comment is that several of the analyses appear to have been performed for the sake of doing the analysis without consideration for the meaning. The manuscript would also benefit from revision to improve clarity by improving the figures and legends, by providing additional details of the methods and some revision to improve the English.

The Methods and Figure legends are generally sufficient but I have some specific questions that should be addressed to provide better clarity and transparency of the analyses.

- For the hormone treatments, please confirm that 100 mM and 1 mM concentrations were used (i.e. milli and not micromolar), which are extremely high concentrations. Also, please describe the controls in more detail. Were controls treated with an appropriate mock solution?

- The statistics used for the qPCR experiment are poorly described. There appears to be a mixture of statistics applied (two-way ANOVA?), but neither are explained very well in the legend and not at all in the Methods. This makes it difficult to evaluate whether appropriate statistics were used.

- The replication for the qPCR analysis could be better explained. In the methods, it is suggested that the experiment was replicated three times and that each samples was analyzed in triplicate, but the meaning of the triplication is unclear. Does this mean that each experiment was performed with three biological replicates (i.e. three separate plants used for RNA extractions to produce three separate RNA samples) or that each sample was run in triplicate for qPCR as technical replicates? If the triplicates are technical replicates, does the data reported in Fig 3 show variation (SE) from technical replicates or the separate experiments?

- There are two separate correlation analyses performed for the expression analysis. The second analysis, pairwise correlation of the treatments for each gene, is relatively clear. However, the first correlation analysis is confusing. Is this actually a linear regression of the time course with a correlation reported for the regression? If so, what is the purpose of the regression and what is the biological relevance?

Several of the figures or tables appear to be redundant or to present information that is not relevant. For example, is Fig 1e necessary? It appears to be a similar analysis as Fig 1d, but with less information since the relevance of the motifs are unknown. There are also several analyses that do not appear to be very informative. Unless the analysis contributes some relevant knowledge, I would recommend removing the analysis from the study. Specifically, the biological relevance is unclear for the sequence logo of Fig 1f and the analysis of the protein secondary structure. Perhaps the sequence logo analysis would have more meaning if the conserved sites were compared with the in silico docking analysis (Fig 2). Otherwise, I would suggest removing Fig 1f and the description of the secondary structure. Also, Tables 1 and 2 provide general information that might be better suited as supplemental tables.

The Figures and Tables could also be revised for clarity. Some specific suggestions include:

- Verify that the font size is legible (Fig 1a and Fig 2 fonts are too small). The right side of Fig 2 is also cropped, suggesting that the figure should be resized.

- Some Figures and Tables could be reorganized. For example, Fig 3a-e would probably fit better as panels in Fig 2.

- The Figures are described out of order. Please revise such that the figure numbering matches the order in which they are referred to in the text.

- The heatmap legend for Fig 1b does not appear to match the colors in the figure panel. What do these colors mean?

The quality of the English is acceptable but the manuscript would generally benefit from some revision. Some general aspects that should be addressed are below:

- The manuscript uses language that suggests a degree of certainty (e.g., confirm, indicate, demonstrate) but since many of the results are shown only by in silico analysis, statements should reflect a degree of uncertainty (e.g., suggest, or use “may” with words like indicate or demonstrate) or further experiments performed to verify the in silico analysis.

- Revise for accuracy since some statements are confusing or may not be accurate. For example, what is meant by “antimicrobial pathogens” (line 46)? Aren’t most (or all) pathogen considered to be microbes? Also, it is stated that phytoalexins are a relatively small group of molecules in peanuts that belong to the stilbene family." Is it certain that there are no other phytoalexins in peanut?

Minor comments:

- STS4 is 100 aa shorter than the other predicted proteins, is it certain to be full length? Was there any attempt, such as RACE, to verify that the STSs are full length? Perhaps reporting 100% coverage with other full length STSs would be helpful.

Author Response

Response to Reviewer 2 Comments

Point 1: For the hormone treatments, please confirm that 100 mM and 1 mM concentrations were used (i.e. milli and not micromolar), which are extremely high concentrations. Also, please describe the controls in more detail. Were controls treated with an appropriate mock solution?

Response 1: It has been updated in the revised manuscript. Please refer to the lines 426 – 432.

Point 2: The statistics used for the qPCR experiment are poorly described. There appears to be a mixture of statistics applied (two-way ANOVA?), but neither are explained very well in the legend and not at all in the Methods. This makes it difficult to evaluate whether appropriate statistics were used.

Response 2: It has been updated in the revised manuscript. Please refer to the material and methods section 4.10. Statistical Analysis in the revised manuscript.

Point 3: The replication for the qPCR analysis could be better explained. In the methods, it is suggested that the experiment was replicated three times and that each samples was analyzed in triplicate, but the meaning of the triplication is unclear. Does this mean that each experiment was performed with three biological replicates (i.e. three separate plants used for RNA extractions to produce three separate RNA samples) or that each sample was run in triplicate for qPCR as technical replicates? If the triplicates are technical replicates, does the data reported in Fig 3 show variation (SE) from technical replicates or the separate experiments?

Response 3: It has been updated in the revised manuscript. Please refer to the lines 457 – 458 in the revised manuscript.

Point 4: There are two separate correlation analyses performed for the expression analysis. The second analysis, pairwise correlation of the treatments for each gene, is relatively clear. However, the first correlation analysis is confusing. Is this actually a linear regression of the time course with a correlation reported for the regression? If so, what is the purpose of the regression and what is the biological relevance?

Response 4: It has been updated in the revised manuscript.

Point 5: Several of the figures or tables appear to be redundant or to present information that is not relevant. For example, is Fig 1e necessary? It appears to be a similar analysis as Fig 1d, but with less information since the relevance of the motifs are unknown. There are also several analyses that do not appear to be very informative. Unless the analysis contributes some relevant knowledge, I would recommend removing the analysis from the study. Specifically, the biological relevance is unclear for the sequence logo of Fig 1f and the analysis of the protein secondary structure. Perhaps the sequence logo analysis would have more meaning if the conserved sites were compared with the in silico docking analysis (Fig 2). Otherwise, I would suggest removing Fig 1f and the description of the secondary structure. Also, Tables 1 and 2 provide general information that might be better suited as supplemental tables.

Response 5: It has been updated in the revised manuscript. Please refer to Figure 1, and supplementary file Table S2, S3.

The Figures and Tables could also be revised for clarity. Some specific suggestions include:

Point 6: Verify that the font size is legible (Fig 1a and Fig 2 fonts are too small). The right side of Fig 2 is also cropped, suggesting that the figure should be resized.

Response 6: It has been updated in the revised manuscript. Please refer to the Figure 2, Figure 3 in the revised manuscript.

Point 7: Some Figures and Tables could be reorganized. For example, Fig 3a-e would probably fit better as panels in Fig 2.

Response 7: It has been updated in the revised manuscript. Please refer to the supplementary file Fig S10-S14.

Point 8: The Figures are described out of order. Please revise such that the figure numbering matches the order in which they are referred to in the text.

Response 8: It has been updated in the revised manuscript. Please refer to the Figure 1, 2, 3, 4, 5 in the revised manuscript.

Point 9: The heatmap legend for Fig 1b does not appear to match the colors in the figure panel. What do these colors mean?

Response 9: It as been updated in the revised manuscript. Please refer to the Figure 3 and line 19–196 in the revised manuscript.

The quality of the English is acceptable but the manuscript would generally benefit from some revision. Some general aspects that should be addressed are below:

Point 10: The manuscript uses language that suggests a degree of certainty (e.g., confirm, indicate, demonstrate) but since many of the results are shown only by in silico analysis, statements should reflect a degree of uncertainty (e.g., suggest, or use “may” with words like indicate or demonstrate) or further experiments performed to verify the in silico analysis.

Response 10: It has been updated in the revised manuscript. Please refer to the lines 184,190, 206 in the revised manuscript.

Point 11: Revise for accuracy since some statements are confusing or may not be accurate. For example, what is meant by “antimicrobial pathogens” (line 46)? Aren’t most (or all) pathogen considered to be microbes? Also, it is stated that phytoalexins are a relatively small group of molecules in peanuts that belong to the stilbene family." Is it certain that there are no other phytoalexins in peanut?

Response 11: It has been updated in the revised manuscript. Please refer to line 47. The phytoalexin group is limited to stilbene related compounds in peanut and contains resveratrol and piceid only.

Minor comments:

Point 12: STS4 is 100 aa shorter than the other predicted proteins, is it certain to be full length? Was there any attempt, such as RACE, to verify that the STSs are full length? Perhaps reporting 100% coverage with other full-length STSs would be helpful.

Response 12: We performed insilico analysis to predict the length of the protein. Morevoer, the full length protein was predicted by the prescence of specific functional domains within the given region. Please refer to the Fig 1a. At this stage we did not perform RACE for verification of the full length sequences.

Reviewer 3 Report

Authors studied the peanut genome to identify more members of  the STS gene family and investigate the individual expression of STS genes to hormonal  treatment. In this study, five AhSTSs genes were identified and reported for the first time  from the Peanut reference genome along with their isolation and sequencing. . The manuscript is well structured and well discussed. However, some points should be checked and corrected before its acceptance in this journal. 

Therefore, according to my comments, I recommended the publication of the paper after minor revision.

  • Please speculate on the results. The discussion must improve.
  • In Conclusion, the authors should add the significance of this research and its potential practical application.
  • The MS English needs to be improved. The article's English must be carefully checked for grammatical errors.

Author Response

Response to Reviewer 3 Comments

Point 1: Please speculate on the results. The discussion must improve.

Response 1: It has been updated in the revised manuscript. Please refer to the results and discussion section in the revised manuscript.

Point 2: In Conclusion, the authors should add the significance of this research and its potential practical application.

Response 2: It has been updated in the revised manuscript. Please refer to the line 477 – 479.

Point 3: The MS English needs to be improved. The article's English must be carefully checked for grammatical errors.

Response 3: The manuscript has been carefully checked for grammatical errors.

Round 2

Reviewer 1 Report

Manuscript plants-1725377-R1, Zunera Iqbal et al., “Identification and Expression Analysis of Stilbene Synthase Genes in Arachis hypogaea in Response to Methyl Jasmonate and Salicylic Acid Induction”, Plants. This paper describes identification and expression analysis of stilbene synthase genes in peanut Arachis hypogaea after methyl jasmonate or salicylic acid treatment. Therefore, the topic of this manuscript (Ms) is interesting and relevant for Plants. Authors improved the Ms text compare with previously version. However, I have several critical remarks to the Ms. Ms needs Major Revisions.

1) Response: Yes, we agree that quantification of stilbene content should be conducted to correlate expression of stilbene gene with the stilbene content. However, due to lack of resources and facilities we could not perform quantification of stilbene. So, our study is limited to the expression of stilbene synthase genes to MeJA and SA treatment.”

- This is a significant drawback of this work. The authors should use HPLC analysis for stilbene detection. It is clear that this cannot be done quickly, then the authors should answer the following comments and questions in detail.

2) Response: It has been updated in the revised manuscript. Please refer to the line 427 – 429 in revised manuscript.”

a) “while SA concentration was decided on the effective concentration for expression from literature” – Authors should include the references.

b) “The solution of MeJA and SA were prepared in distilled water. Ethanol was used to dissolve MeJA and SA in the water to form the hormonal solution.” – It is not clear to me. What the final volume of ethanol was in the working solution?

c) “Hormonal solution was then sprayed on the leaves of the peanut plant” – It is not clear to me. What is the volume of the hormonal solution and for how many plants were sprayed? What was the condition of the plants (age, size)?

3) Response: It has been updated in the revised manuscript.”

- The described problem remained in the new manuscript (Ms), for example, high errors in the Figure 5 (b, 3 h; e, control).

4) Response: The quality of figures have been improved in the revised manuscript. Please refer to the Figures 1, 2, 3 and 4.”

- The described problem remained in the new Ms, for example, Fig. 1, 2, 4, 5 – please, pay special attention to the font. It is difficult to read the inscriptions.

5) ResponseTwo internal control are not needed here. The actin was used as per available literature and was worth to go for the expression analysis of MeJA and SA.”

- “Two internal control are not needed here” – Please, explain.

6) Why did the authors remove “Golden peanut variety” from the text of the Ms?

Round 3

Reviewer 1 Report

Major:

1) Line 496-499: why did the authors use 150 ml in one case (MeJa) and 200 ml in the second (SA)?

2) Line 498-500: “1mM SA stock solution was prepared by dissolving 138.121 mg of SA in warm 200 ml dH20 with 10m ml absolute ethanol.”

a) I do not agree, if the authors used 138.12mg per 200 ml, then it will be a 5 mM SA solution, not 1 mM.

b) Why authors used “stock solution”? Stock solutions are used to save preparation time, conserve materials, reduce storage space, and improve the accuracy with which working lower concentration solutions are prepared. As I understand, the authors used the obtained 150 and 200 ml for plant treatments? Thus, it is the working solutions.

c) Correct “138.121” to “138.12”.

d) “10m ml absolute ethanol” correct to “10 ml absolute ethanol”.

3) Line 500-501: “Hormonal solution was then sprayed two times individually on the leaves of the 15 peanut plants for each hormone”

a) Age of the used plants?

b) “then sprayed two times” – immediately 2 times 150 ml (MeJa) or 200 ml (SA)? Or after some time period? As a result, 300 (for MeJa) and 400 (for SA) ml were used?

4) “Point 4: “Two internal control are not needed here” – Please, explain.

Response 4: For analyzing relative gene expression, Livak method has been used. This method requires only one housekeeping gene for normalizing the expression while other second normalization is done through control. Yes, you are right before performing experiment, we need to check various housekeeping genes that would be best suited for the analysis. Moreover, our group has good experience while working with actin. After carefully checking various pentameters and literature review actin has been chosen for final analysis as housekeeping gene.”

Livak method was published in 2001. After that, many papers were published using this approach and as a result, scientists concluded that there is no universal housekeeping gene. The expression of housekeeping genes may be affected by the tissue or the used stress. Therefore, in order to get more accurate data, scientists concluded that it is better to use at least 2 controls that relate to different systems in the cell. For the future experiments, authors should use at least 2 internal controls in real time PCR

5) Authors should present the manuscript with only last changes. Changes that will be made in response to the above comments. This will make it easier to reread the manuscript.
